# Efficacy of Feed Additive Containing Bentonite and Enzymatically Hydrolyzed Yeast on Intestinal Health and Growth of Newly Weaned Pigs under Chronic Dietary Challenges of Fumonisin and Aflatoxin

**DOI:** 10.3390/toxins15070433

**Published:** 2023-06-30

**Authors:** Zixiao Deng, Ki Beom Jang, Sangita Jalukar, Xiangwei Du, Sung Woo Kim

**Affiliations:** 1Department of Animal Science, North Carolina State University, Raleigh, NC 27695, USA; zdeng7@ncsu.edu (Z.D.); kjang@ncsu.edu (K.B.J.); 2Arm & Hammer Animal and Food Production, Church & Dwight Co., Inc., Ewing, NJ 02628, USA; sangita.jalukar@churchdwight.com; 3College of Veterinary Medicine, University of Missouri, Columbia, MO 65211, USA; x.du@missouri.edu

**Keywords:** bentonite, enzymatically hydrolyzed yeast, intestinal health, mycotoxin, nursery pigs

## Abstract

This study aimed to investigate the efficacy of a feed additive containing bentonite and enzymatically hydrolyzed yeast on the intestinal health and growth of newly weaned pigs under chronic dietary exposure to fumonisin and aflatoxin. Newly weaned pigs were randomly allotted to one of four possible treatments: a control diet of conventional corn; a diet of corn contaminated with fumonisin and aflatoxin; a diet of mycotoxin-contaminated corn with 0.2% of feed additive; and a diet of mycotoxin contaminated corn with 0.4% of feed additive. We observed lower average weight gain and average daily feed intake in pigs that were fed only mycotoxin-contaminated corn compared to the control group. Feed additive supplementation linearly increased both average weight gain and feed intake, as well as tumor necrosis factor-alpha. In the jejunum, there was an observed decrease in immunoglobulin A and an increase in claudin-1. Additionally, feed additive supplementation increased the villus height to crypt depth ratio compared to the control. In conclusion, feed additives containing bentonite and enzymatically hydrolyzed yeast could mitigate the detrimental effects of mycotoxins on the growth performance of newly weaned pigs by improving intestinal integrity and positively modulating immune response.

## 1. Introduction

Mycotoxins are secondary metabolites produced by fungi in nature. The common genera producing mycotoxins are *Aspergillus*, *Fusarium*, and *Penicillum* [1]. Mycotoxins can cause various toxic effects depending on the level, type of mycotoxins, animal species, sex, environment, and health status [2,3]. Fumonisins (FUM) are mainly produced by *Fusarium* spp. and are identified into four main groups, including fumonisin A (FA), B, C, and P series [4]. Fumonisin B (FB) series, including FB1, FB2, and FB3, are the most prevalent in nature and crucial in terms of toxicology [5]. Aflatoxins (AF) are mainly produced by *Aspergillus* spp. and are identified into aflatoxin B1 (AFB1), AFB2, AFG1, and AFG2 [6]. Both FUM and AF are commonly discovered in cereal grains and feedstuffs, negatively influencing the health and growth of animals [3,7].

Pigs are considered one of the most sensitive species to mycotoxins, especially at a young age [3,7,8]. The FB has been indicated to impair the intestinal barrier and immune response and then reduce feed intake and growth performance of pigs [9]. The mechanism of action of FB was shown that due to the similar structure to cellular sphingolipids, it could interfere with the normal metabolism of sphingolipids by blocking the activity of ceramide synthase, Then it increased oxidative stress, apoptosis, and cytotoxicity [10]. The pigs that consumed aflatoxin-contaminated grains have been suggested to reduce growth performance, impair liver and kidney function, and impair the cellular humoral immune system [11,12]. The mechanism of action of AF could be associated with gene transcription and protein synthesis, which affects regular normal metabolic activity and regulation [13].

Bentonite is a type of clay rock that is primarily composed of smectite [14]. The commercial value of bentonite is derived from the exceptional properties of smectite, including its submicrometer crystal size, sheet-like structure, significant surface area, negative charge, and the ability to exchange cations [15]. Due to its characteristics, it is widely used in the feed industry to mitigate the effect of mycotoxins on animals as an adsorbent [16]. Enzymatically hydrolyzed yeast (EHY) is produced by breaking down yeast through hydrolysis, resulting in a mixture of yeast extract and yeast cell walls [17]. Yeast is widely used in swine diets as a feed additive due to its function to positively modulate the immune system, maintain a balanced intestinal environment, and be a potential alternative of antimicrobial growth promoters for pigs [18,19,20]. In addition, previous studies have indicated that the inclusion of yeast cell walls in nursery diets has the potential to reduce the harmful impacts of mycotoxins on pigs [21,22,23].

Therefore, it was hypothesized that the inclusion of the feed additive containing bentonite and EHY in diets would mitigate the detrimental impacts of naturally co-occurring FUM and AF on the intestinal health and growth performance of nursery pigs. To test the hypothesis, the objective of this study was to investigate the effects of the feed additive containing bentonite and EHY on intestinal health and growth performance of nursery pigs under chronic dietary challenges of naturally co-occurring FUM and AF.

## 2. Results

### 2.1. Growth Performance and Fecal Score

The initial BW of nursery pigs at the beginning was 6.2 ± 0.3 kg and there was no difference among treatments (Table 1). The PC decreased (*p* < 0.05) the BW of pigs on d 11 and d 22 after weaning and tended to decrease (*p* = 0.051) the BW of pigs on d 32 compared with NC. Increasing the supplementation of feed additive tended to linearly increase the BW of pigs fed mycotoxin-contaminated diets on d 22 (*p* = 0.080) and d 32 (*p* = 0.056).

The PC decreased (*p* < 0.05) the ADG of pigs during phase 1 and tended to decrease (*p* = 0.050) the ADG of pigs during the overall period compared with NC. Increasing the supplementation of feed additive tended to linearly increase the ADG of pigs fed mycotoxin-contaminated diets during phase 1 (*p* = 0.085), phase 3 (*p* = 0.097), and overall period (*p* = 0.060). The PC decreased (*p* < 0.05) the ADFI of pigs compared with NC during phase 1, phase 2, phase 3, and the overall period. Increasing the supplementation of feed additive tended to linearly increase the ADFI of pigs fed mycotoxin-contaminated diets during phase 3 (*p* = 0.079) and overall period (*p* = 0.066). The BG (BG1 + BG2) tended to decrease (*p* = 0.063) the ADFI of pigs in phase 1. The PC decreased (*p* < 0.05) G:F of pigs during phase 1, whereas it tended to increase (*p* = 0.081) the G:F of pigs during phase 3 compared with NC. Increasing the supplementation of feed additive tended to linearly increase (*p* = 0.064) the G:F of pigs fed mycotoxin-contaminated diets during phase 1. The BG (BG1 + BG2) tended to increase (*p* = 0.064) the G:F of pigs in phase 2 compared with NC. The PC tended to increase (*p* = 0.071) fecal score of pigs in phase 1 (Table 2). Increasing the supplementation of feed additive linearly decreased (*p* < 0.05) the fecal score of pigs fed mycotoxin-contaminated diets in phase 1 and phase 2.

### 2.2. Serum Sphinganine and Sphingosine

The PC increased (*p* < 0.05) the SA:SO in the serum of pigs. The increasing supplementation of feed additives in the mycotoxin diets did not alter the SA:SO in the serum of pigs fed mycotoxin-contaminated diets (Table 3).

### 2.3. Immune and Oxidative Stress Status

Increasing the supplementation of feed additive tended to linearly increase (*p* = 0.050) TNF-α in jejunal mucosa of pigs fed mycotoxin-contaminated diets, whereas it tended to linearly decrease (*p* = 0.088) IgA in jejunal mucosa of pigs fed mycotoxin-contaminated diets (Table 4).

### 2.4. Gene Expression of Tight Junction Proteins

Increasing the supplementation of feed additive in the mycotoxin diets tended to linearly increase (*p* = 0.082) the expression of claudin-1 in the jejunum of pigs fed mycotoxin-contaminated diets (Figure 1). The BG (BG1 + BG2) tended to increase (*p* = 0.080) the expression of claudin-1 in the jejunum of pigs compared with NC.

### 2.5. Histology and Immunohistochemistry

The BG (BG1 + BG2) tended to increase villus height (*p* = 0.066) and VH:CD (*p* = 0.098) in the jejunum of pigs compared with NC (Table 5).

## 3. Discussion

Fumonisin (FUM) and aflatoxin (AF) are two common mycotoxins in naturally contaminated grains that lead to detrimental impacts on the health and growth of pigs [24]. The Food and Drug Administration (FDA) has established guidelines specifying that the levels of FUM and AF in swine diets should not exceed 10 mg/kg and 0.2 mg/kg, respectively [25,26]. A previous study has shown that the inclusion level of FUM at 21.9 mg/kg in the diet can result in negative effects on the growth performance of nursery pigs [27]. The diets containing 0.25 mg/kg AF could reduce the ADG of nursery pigs during the first three weeks [28]. Furthermore, previous studies have also indicated that mycotoxins may have synergistic toxic effects, such as impaired immune health, systemic inflammation, and hepatic histopathology changes, that can cause more damage to the growth performance when multiple mycotoxins are ingested [12,29,30]. In this study, two mycotoxins were included in the experimental diets since the co-occurrence of mycotoxins was highly common in nature [31]. This study was to provide a practical insight into the effects of multiple mycotoxins on nursery pigs under regular feeding conditions. The conventional corn used in this study was not free of mycotoxins. It contained FUM resulting in about 1 mg/kg FUM in the diet. Mycotoxin contamination in conventional corn was not avoidable even though the impact of FUM at 1 mg/kg feed on the growth and health of nursery pigs is insignificant [3,25]. 

The combination of 8 mg/kg FUM and 0.19 mg/kg AF in the diets led to an impairment in the growth performance of nursery pigs compared to the NC treatment. Holanda et al. [32] demonstrated that feeding 8 mg/kg FUM and 0.19 mg/kg AF negatively affected the growth performance of nursery pigs. However, unlike this study showing the detrimental impacts of FUM and AF observed at the beginning of the feeding period, in the previous study [32], the negative effects of FUM and AF on growth performance were observed several days after feeding. It is speculated that the initial body weights of pigs between these two studies were different and thus the mycotoxin susceptibility of pigs would be different at the beginning. In general, mycotoxins can cause more severe impacts for post-weaned pigs than older pigs due to the immature immune system and weaning stress [21,33]. Weaver et al. [33] demonstrated the importance of the intestinal health status of newly weaned pigs handling mycotoxin challenges. Correspondingly, the fecal score of pigs was increased in phase 1 with the inclusion of FUM and AF in the diets. This result was consistent with a previous study suggesting that nursery pigs fed AFB1 contaminated diet increased diarrhea incidence [34]. This increased diarrhea can be explained by the impaired immune system of pigs when exposed to FUM and AF [35,36]. Since the immune system is primarily responsible for combating external pathogens, any damage to it may lead to a higher susceptibility to infections, resulting in an increased fecal score in nursery pigs. After feeding for two weeks, the effects of mycotoxins on feed efficiency, ADG, and fecal score were diminished in this study. Similar results were previously shown by Holanda et al. [32]. In addition, Swamy et al. [37] found that nursery pigs would recover weight gains after feeding two weeks of *Fusarium* mycotoxins contaminated diets. This outcome also indicates that pigs may become less susceptible to mycotoxins as pigs mature.

With the increasing supplementation of the feed additive, the detrimental effects of mycotoxins on growth performance and fecal score were alleviated in this study. As the primary functional compound in the feed additive, yeast cell walls have the capacity to adsorb *Fusarium* mycotoxin through hydrogen and ionic binding or hydrophobic interactions [38]. Bentonites, as the secondary functional compound in the feed additive, also have a high capacity for adsorbing mycotoxins such as AF and FUM [39,40]. The improved growth performance and fecal score observed in this study can be attributed to the increased mycotoxin adsorbent in the feed additive, which can effectively adsorb more mycotoxins from the diets. In addition to their ability to adsorb mycotoxins, the improved intestinal health resulting from the use of bentonite and EHY could be another contributing factor. Previous studies have indicated that clay or yeast culture can alleviate diarrhea, stimulate the immune system, or positively modulate the intestinal environment of pigs [18,41]. The improved intestinal health might contribute to enhanced growth performance and decreased fecal score. However, this study did not assess the fecal output of mycotoxins which can be an indicator of the mycotoxin binding capacity of feed additives used in this study. 

Fumonisin is generally considered neurotoxic, hepatoxic, and nephrotoxic for animals, which can result in the change of oxidative stress, apoptosis, and modulation of cytokine expression, such as TNF-α [10]. The ratio of SA to SO could be an early biomarker for pigs that consumed FUM-contaminated diets since fumonisins could specifically inhibit SO and SA *N*-acyltransferases [42]. In this study, the ratio of SA to SO increased in PC treatment compared to NC treatment, which was in accordance with a previous study [27]. However, the inclusion of feed additives did not alleviate the increased ratio of SA to SO, which might be related to the weak adsorption to FUM or increased feed intake in BG treatments. In addition, researchers indicated that AFB1 could interfere with the normal function of dendritic cells and then inhibit the T-cell proliferation thus affecting normal immune response [43]. In this study, however, no difference in immune and oxidative stress between NC and PC was observed. Even though several studies showed that FUM or AF could alter the inflammation and immune status of pigs [12,35,44], there were some studies showing inconsistencies. Weaver et al. [45] found that feeding AF and DON did not alter the cytokine TNF-α. Taranu et al. [36] and Meissonnier et al. [46] showed that the inclusion of AFB1 in the diet did not affect the IgG and IgA levels. The different results might be explained by the level of mycotoxins in the diets and the health status of pigs. Interestingly, increasing the supplementation of feed additives containing bentonite and EHY increased the pro-inflammatory cytokines. The result was consistent with previous studies that the supplementation of nucleotide-rich yeast extract upregulated the pro-inflammatory cytokines, such as IL-1β, IL-6, and TNF-α, in pigs [47]. Yeast and yeast derivatives have been indicated to stimulate the release of TNF-α from macrophages and β-glucan in yeast cell walls could improve the functionality of macrophages and neutrophils [48]. The modulation in cytokines and immune cells caused by yeast and yeast derivatives could activate the immune system in livestock, which could help animals to mitigate the negative effects related to stress, such as exposure to pathogens, environmental changes, or other factors [20]. However, increasing the supplementation of feed additives in the diets decreased the level of IgA in the intestine. This finding is consistent with a previous study that investigated the effects of feeding pigs a diet supplemented with 0.2% clay additive during chronic exposure to aflatoxin and deoxynivalenol [45]. Due to the antibacterial effects of clay [49,50], it is therefore speculated that increasing the supplementation of feed additives may reduce the need for animals to produce immunoglobulins to combat potential pathogens in the intestine. It was also indicated that IgA was the most effective immunoglobulin against *Escherichia coli* by inhibiting the adsorption or blocking the binding to receptors on epithelial cells [51]. Nevertheless, further research that how feed additive containing bentonite and EHY affect intestinal microbiota under the challenge of FUM and AF is needed to fully explain the immune response.

The intestinal barrier consisting of epithelial cells and intercellular junction proteins plays a role in preventing the invasion of toxic compounds and potential parthenogens [52,53]. Tight junction proteins are composed of membrane-bound proteins (occludin and claudins) and scaffolding proteins (junctional adhesion molecule and zonula occludens, and adaptors) [54]. In this study, the inclusion of FUM and AF in the nursery diets did not affect the expression of tight junction proteins, which was not in agreement with the result in a previous study suggesting that FUM or AF could impair the tight junction proteins [55]. However, other researchers found that the inclusion of 0.18 mg/kg AFB1, 1 mg/kg DON, and 9 mg/kg FB1 in the diets did not affect tight junction proteins in the jejunum [21], which was consistent with the result in this study. It could be explained by the relatively low concentration of mycotoxins in the diets and less susceptibility of pigs to aging. With increasing the supplementation of the feed additive, the expression of claudin-1 (CL-1) linearly increased in this study. This was in accordance with a previous study suggesting that the supplementation of mycotoxin mitigation product containing bentonite could increase the expression of CL-1 in pigs [56]. Furthermore, yeast culture has been indicated to improve the intestinal tight junction proteins through pathways of nucleotide-binding oligomerization domain protein 1 (NOD-1) and nuclear transcription factor κB (NF-κB) in pigs [57]. The improved tight junction protein could help animals properly absorb nutrients, water, and electrolytes and against mycotoxins and potential pathogens [58].

Weaning can impact intestinal morphology and thus impair the absorptive capacity of nursery pigs [59]. Therefore, the restoration of intestinal morphology after weaning would be very essential for nursery pigs. It was reported that exposure to FUM or AF could damage the intestinal morphology of animals [60,61]. However, other researchers showed that FUM or AF did not affect intestinal morphology or increase the villus height [21,62]. In this study, no differences in intestinal morphology between NC and PC were observed. It is speculated that the effects of mycotoxins on intestinal morphology depend on the concentration of mycotoxins in the diets and the exposure time. As pigs mature, they may become less susceptible to the negative effects of low levels of FUM and AF, and their bodies may begin to recover from the damage caused by earlier exposure. The supplementation of feed additive increased the villus height and VH:CD of pigs under the mycotoxin challenge in this study. This result was consistent with the previous study suggesting that feeding copper-bearing montmorillonite could improve the gastrointestinal morphology in weanling pigs [63]. In addition, studies have reported that yeast culture and autolyzed yeast supplementation could improve the intestinal morphology in newly weaned pigs [19,64]. The improved intestinal morphology in pigs could be an indicator related to nutrients absorption from feeds [65].

## 4. Conclusions

The presence of FUM and AF in the diet reduced the growth performance of nursery pigs and the supplementation of feed additive containing bentonite and EHY could mitigate the detrimental effects of mycotoxins on the growth performance of nursery pigs, which were contributed by improved the integrity, positively modulated immune response, and enhanced morphology in the jejunum.

## 5. Materials and Methods

### 5.1. Experimental Design, Animals, and Diets

The North Carolina State University Animal Care and Use Committee (Raleigh, NC, USA) conducted a thorough review and approved the protocol employed in the study. The experiment was carried out at the North Carolina State University Metabolism Educational Unit (Raleigh, NC, USA).

Forty-eight newly weaned pigs (24 barrows and 24 gilts; initial BW 6.2 ± 0.3 kg) were randomly allotted to 4 treatment groups based on randomized complete block design with BW (heavy and light) and sex (barrows and gilts) as blocks. Pigs were assigned to individual pens and each treatment group had 12 replicates. The treatment groups were as follows (1) NC: a diet with conventional corn, (2) PC: NC with naturally contaminated corn (fumonisin at 8 mg/kg and aflatoxin at 0.19 mg/kg); (3) BG1: PC with 0.2% of feed additive (BG-MAX, Church & Dwight, Trenton, NJ, USA); and (4) BG2: PC with 0.4% of feed additive. The feed additive contains 60 to 100% processed grain by-products, 10 to 30% bentonite, 1 to 5% EHY, and 0.1 to 1% white mineral oil. The experiment lasted for 32 d and was divided into 3 phases: phase 1 (weaning to d 11), phase 2 (d 11 to 22), and phase 3 (d 22 to 32). The length of each phase was determined by the average body weight of pigs as suggested by NRC (2012) [66]. The pigs had free access to both feed and water throughout the duration of the experiment. The ADG, ADFI, and G:F were calculated using measurements of feed intake and body weight taken at the beginning and end of the experiment. The fecal score of the pigs was evaluated and recorded using a 5-point scale, ranging from very firm stool (1), normal firm stool (2), moderately loose stool (3), loose and watery stool (4) to very watery stool (5).

Experimental diets (Table 6) were formulated to meet or exceed the nutrient requirements suggested by NRC (2012). All experimental diets were sampled (total 2 kg per treatment diet) from 9 different locations in mixing batch, and then two subsamples (300 g) of each were sent to the North Carolina Department of Agriculture (Raleigh, NC, USA) for composition analysis. To determine the FUM concentration, the commercial kit (#8832, Neogen, Lansing, MI, USA) was used. All the processes followed the instruction manual. To determine the AF concentration, diet samples were sent to the North Carolina Department of Agriculture (Raleigh, NC, USA) for the measurements.

### 5.2. Sample Collection and Processing

On d 28, pigs were placed in a recumbent position on a V-shaped table to restrict their movement, and blood samples of each pig were collected (10 mL) using vacutainer tubes (Becton Dickinson Vacutainer Systems, Franklin Lakes, NJ, USA) and needles (0.8 mm × 32 mm needles, Eclipse, Becton Dickinson Vacutainer Systems) by puncturing the jugular vein. After clotting at 4 °C for 4 h, the serum samples were centrifuged at 3000× *g* at 4 °C for 15 min (5811F, Eppendorf, Hamburg, HH, Germany). Thereafter, the serum was transferred into 1.5 mL tubes and stored at −80 °C for subsequent analysis. In order to prevent prolonged stress from blood sampling, the blood samples were collected 4 days before the end of the feeding period.

At the end of the experiment, all pigs were euthanized through the use of a captive bolt, followed by exsanguination to collect the samples. Mid jejunal tissue was obtained and washed with 0.9% saline solution. The mid-jejunal mucosal tissues were sampled by gently scraping the mucosal layer with a glass microscope slide. Two pieces of mid-jejunal segments (5 cm) were also collected. One piece was fixed with 10% buffered formaldehyde in a 50 mL falcon tube. Another piece was stored at −80 °C until further analyses.

### 5.3. Serum Sphinganine and Sphingosine 

The serum samples were forwarded to the Veterinary Medical Diagnostic Laboratory at the University of Missouri (Columbia, MO, USA) for analysis of the sphinganine (SA) to sphingosine (SO) ratio. The assessment was conducted using high-performance liquid chromatography (HPLC) with fluorescence detection, employing a customized approach [67]. In the beginning, 0.5 mL of each serum sample was carefully transferred into 15 mL polypropylene tubes, after which 2.0 mL of methanolic 0.125 M KOH and 0.5 mL of chloroform were added. The samples were mixed by vertexing and incubated in a water bath at 37 °C for 1 h to facilitate the extraction of sphinganine and sphingosine. Thereafter, 2.0 mL of chloroform, 2.0 mL of alkaline water, and 0.5 mL of 2 N ammonium hydroxide were added to the samples for further extraction. The samples were vortexed, shaken for 10 min, and subsequently centrifuged at 3300 rpm (1920 g) for 15 min. The lower chloroform layer (2 mL) was carefully transferred to polypropylene tubes containing 4 mL of alkaline water. The tubes were vortexed and shaken for 10 min and centrifuged for 10 min at 3300 rpm for 15 min. The lower chloroform layers (1.5 mL) were transferred into 5 mL vials and evaporated to dryness. 

The residues were reconstituted in 1.2 mL methanol, and sonicated for 15 min. Each reconstituted solution (600 μL) was transferred to autosampler vials and was derivatized with 300 μL o-Phthaldialdehyde (OPA) reagent, vortexed, and loaded into an autosampler for analysis by HPLC individually. The HPLC system employed for analysis consisted of several components, including a Hitachi Model L-7100 pump, a Hitachi Model L-7485 fluorescence detector (Ex-230 nm; Em-430 nm), a Hitachi Model L-7200 autosampler, a data acquisition interface (Hitachi D-7000, Tokyo, Japan), and ConcertChrom software. For separation, a Hypersil 250 × 4.6 C18 BDS 5 μm column (Phenomenex, Torrance, CA, USA) with a C_18_ SecurityGuard precolumn (Phenomenex) was utilized, with a mobile phase composed of methanol and K_2_HPO_4_ buffer (5 mM, pH 7.0) in a 95:5 ratio. A flow rate of 1.0 mL/min was maintained during the analysis.

### 5.4. Immune and Oxidative Stress Status

To prepare the jejunal mucosa for analysis, a weighed amount (0.5 g) was transferred to a 5 mL tube and mixed with 1 mL of phosphate-buffered saline (PBS, 0.01 M phosphate, 0.0027 M KCl, 0.137 M NaCl, pH 7.4 at 25 °C). The homogenizer (#15340163; Thermo Fisher Scientcific Inc., Waltham, MA, USA) was used to process samples. The processed sample was transferred into a new 2 mL microcentrifuge tube and centrifuged at 14,000× *g* for 15 min. To prepare the samples for further analysis, the supernatant was pipetted into 6 equal aliquots and stored at −80 °C until use. The concentration of total protein, protein carbonyl, tumor necrosis factor-alpha (TNF-α), malondialdehyde (MDA), interleukin 6 (IL-6), immunoglobulin G (IgG), and immunoglobulin A (IgA) was measured by using commercial kits as described by a previous study [68]. The OD value was read by the ELISA plate reader (Synergy HT, BioTek Instruments, Winooski, VT, USA) and software (Gen5 Data Analysis Software, BioTek Instruments). According to the standard curve absorbance and instruction manual, the relevant concentrations were determined. The concentration of total protein, protein carbonyl, TNF-α, MDA, IL-6, IgA, and IgG was measured by Pierce BCA Protein Assay Kit (#23225, Thermo Fisher Scientific), OxiSelect™ Protein Carbonyl ELISA Kit (#STA-310, Cell Biolabs, Inc.; San Diego, CA, USA), Porcine TNF-α Immunoassay Kit (#PTA00, R&D Systems; Minneapolis, MN, USA), OxiSelect™ TBARS MDA Quantitation Assay Kit (#STA-330, Cell Biolabs, Inc.; San Diego, CA, USA), Porcine IL-6 DuoSet ELISA kit (#DY686, R&D Systems; Minneapolis, MN, USA), ELISA kits (E101-102 and E101-104, Bethyl Laboratories, Inc., Montgomery, TX, USA), separately. All procedures were followed by the manufacturer’s protocol.

### 5.5. RNA Extraction and Gene Expression of Tight Junction Proteins

The RNA was extracted from mid-jejunal tissue following the previous study [69]. Frozen mid-jejunal tissue (50 to 100 mg) was mixed in 1 mL tube with pre-cooling Trizol reagent (#15-596-026, Invitrogen, Waltham, MA, USA) then the samples were processed at 4.5 m/s for 30 s two times using a Bead Mill 24 homogenizer (#15-340-163, Thermo Fisher Scientific Inc.). Homogenized samples were centrifuged at 12,000× *g* for 10 min at 4 °C to get supernatants. 200 μL of chloroform (#146543, Thermo Fisher Scientific Inc.) was mixed with the supernatant in a new tube and incubated at room temperature for 10 min. After incubation, the mixed samples were centrifuged to get the aqueous phase and mixed with 200 μL of isopropanol (#B0518327, Acros Organics, Geel, NJ, USA). After 10 min resting, the mixed samples were centrifuged to get the sediment and then mixed with 75% ethanol. The mixed samples were centrifuged to remove the supernatants and then mixed with 40 μL DEPC water. The RevertAid First Strand cDNA Synthesis kit (#01299151, Thermo Fisher Scientific Inc.) was used to revert the extracted RNA into cDNA. All the procedures followed the manufacturer’s instructions. The CFX Connect Real-Time PCR Detection System (BioRad, Hercules, CA, USA) and Maxima SYBR Green/ROX qPCR Master Mix (#01292815, Thermo Fisher Scientific Inc.) was used for quantitative RT-PCR (qRT-PCR). The primers used for the tight junction proteins are listed in Table 7 and were synthesized by a commercial company (Millipore Sigma, Burlington, MA, USA). Delta–delta Ct values were calculated to get a relative expression of each target gene.

### 5.6. Histology and Immunohistochemistry

Mid-jejunal tissues of each pig were used to evaluate intestinal morphology and crypt cell proliferation. The tissues were kept in 10% buffered formaldehyde for 48 h for fixation. Two sections of fixed tissue (approximately 2 mm) were cut, placed in a cassette, and transferred to a 70% ethanol solution. The processed samples were sent to the North Carolina State University Histology Laboratory (College of Veterinary Medicine, Raleigh, NC, USA) for dehydration, embedment, and staining using a Ki-67 assay. The Biocare Intellipath Stainer (Biocare Medical, Pacheco, CA, USA) was used to process the automated Ki-67 staining. To get the proper working concentration, the primary monoclonal antibody of Ki-67 (#ACR325, Biocare Medical) was diluted (1:100) and then incubated for 30 min with processed slides at room temperature. For detection, Vector ImmPress Rabbit polymer was employed. Staining was carried out using chromogen diaminobenzidine (DAB). The microscope Olympus CX31 (Lumenera Corporation, Ottawa, ON, Canada) and software (Infinity 2-2 digital CCD) were used to measure intestinal morphology (villus height, villus width, and crypt depth) at a magnification of 40× following the procedure described by a previous study [72]. Ten complete villi and crypts were chosen to represent the intestinal morphology of each pig. The length of each villus was measured from the tip to the point where it intersects with the crypt, and the depth of each crypt was measured from its base to the point where it intersects with the villus. The villus height was divided by the crypt depth to determine the villus height to crypt depth (VH:CD) ratio. The percentage of Ki-67 positive cells (the marker for proliferating cells in the crypt) was calculated using pictures of 10 complete crypts captured by the Olympus CX31 microscope at a magnification of 100× following the procedure described by Xu et al. [73]. The pictures were cropped and then uploaded to the Image JS tool for analysis. All the procedures were conducted by the same person. 

### 5.7. Statistical Analysis

Data were analyzed by the Mixed procedure of SAS 9.4 (SAS Inst. Inc., Cary, NC, USA). The dietary treatment was considered the main fixed effect, and the initial BW and sex were considered random effects. The experimental unit was a pen. Contrasts were preplanned to determine the effects of mycotoxin mitigation product and dietary supplemental mycotoxin mitigation product for the linear responses. Preplanned contrasts were determined using the Interactive Matrix Language (IML) procedure of SAS to generate coefficients for the equally spaced orthogonal contrasts. The *p* values less than 0.05 were considered statistically significant and between 0.05 and 0.10 were considered tendency.

## Figures and Tables

**Figure 1 toxins-15-00433-f001:**
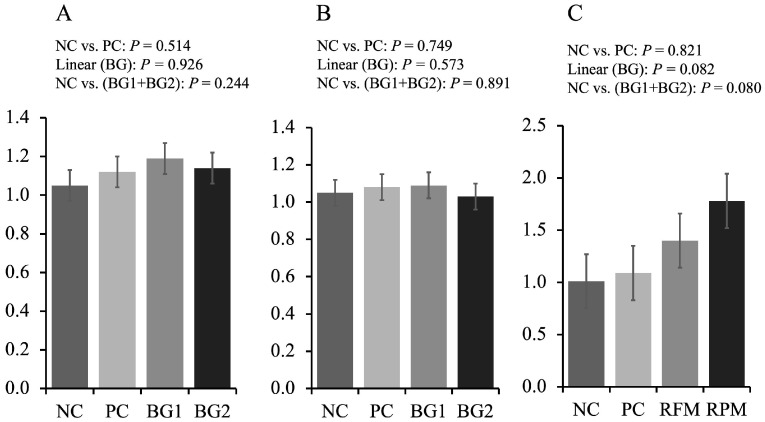
Tight junction proteins of jejunum of nursery pigs fed diets with or without mycotoxins, and with or without feed additive. (**A**) Relative expression of zonula occludens in the jejunum of pigs. (**B**) Relative expression of occludin in the jejunum of pigs. (**C**) Relative expression of claudin-1 in the jejunum of pigs. The *x*-axis is “treatment” and the y-axis is “relative expression”.

**Table 1 toxins-15-00433-t001:** Growth performance of nursery pigs fed diets with or without mycotoxins, and with or without feed additive.

Item	Treatment ^1^					*p* Value		
	NC	PC	BG1	BG2	SEM	NC vs. PC	Linear (BG)	NC vs. BG1 + BG2
BW, kg								
d 0	6.2	6.2	6.2	6.2	0.2	0.899	0.855	0.893
d 11	7.6	6.9	7.1	7.5	0.3	0.041	0.105	0.280
d 22	11.6	10.4	11.0	11.5	0.4	0.045	0.080	0.470
d 32	17.1	15.2	15.8	17.0	0.7	0.051	0.056	0.405
ADG, g/d								
Phase 1	126	63	82	114	21	0.033	0.085	0.262
Phase 2	369	318	357	370	27	0.173	0.169	0.867
Phase 3	542	481	478	545	33	0.184	0.097	0.453
Overall	340	281	300	337	21	0.050	0.060	0.412
ADFI, g/d								
Phase 1	169	117	116	147	21	0.026	0.184	0.063
Phase 2	532	424	472	499	33	0.021	0.101	0.240
Phase 3	838	703	755	819	48	0.051	0.079	0.395
Overall	503	406	438	478	29	0.015	0.066	0.188
G:F								
Phase 1	0.74	0.22	0.62	0.73	0.19	0.036	0.064	0.751
Phase 2	0.70	0.75	0.75	0.74	0.03	0.134	0.889	0.088
Phase 3	0.64	0.69	0.64	0.67	0.02	0.081	0.271	0.675
Overall	0.68	0.69	0.68	0.71	0.02	0.521	0.324	0.340

^1^ NC: control diet formulated with conventional corn; PC: inclusion of 8 mg/kg fumonisins and 0.19 mg/kg aflatoxins supplemented from mycotoxin-contaminated corn; BG1: PC diet with a feed additive (BG-Max, Church and Dwight Co., Inc., Princeton, NJ, USA) at 0.2%; BG2: PC diet with the feed additive at 0.4%.

**Table 2 toxins-15-00433-t002:** Fecal score of nursery pigs fed diets with or without mycotoxins, and with or without feed additive.

Item	Treatment ^1^					*p* Value		
	NC	PC	BG1	BG2	SEM	NC vs. PC	Linear (BG)	NC vs. (BG1 + BG2)
Fecal score								
Phase 1	3.93	4.34	3.85	3.87	0.18	0.071	0.043	0.715
Phase 2	3.15	3.30	3.01	2.86	0.13	0.418	0.004	0.162
Phase 3	3.05	3.03	3.06	2.99	0.04	0.699	0.421	0.550

^1^ NC: control diet formulated with conventional corn; PC: inclusion of 8 mg/kg fumonisins and 0.19 mg/kg aflatoxins supplemented from mycotoxin-contaminated corn; BG1: PC diet with a feed additive (BG-Max, Church and Dwight Co., Inc., Princeton, NJ, USA) at 0.2%; BG2: PC diet with the feed additive at 0.4%.

**Table 3 toxins-15-00433-t003:** Serum sphinganine and sphingosine ratio of nursery pigs fed diets with or without mycotoxins, and with or without feed additive.

Item	Treatment ^1^					*p* Value		
	NC	PC	BG1	BG2	SEM	NC vs. PC	Linear (BG)	NC vs. (BG1 + BG2)
SA:SO ^2^	0.12	0.41	0.42	0.42	0.04	<0.001	0.879	<0.001

^1^ NC: control diet formulated with conventional corn; PC: inclusion of 8 mg/kg fumonisins and 0.19 mg/kg aflatoxins supplemented from mycotoxin-contaminated corn; BG1: PC diet with a feed additive (BG-Max, Church and Dwight Co., Inc., Princeton, NJ, USA) at 0.2%; BG2: PC diet with the feed additive at 0.4%. ^2^ SA:SO, sphinganine to sphingosine ratio.

**Table 4 toxins-15-00433-t004:** Immune and oxidative stress status of nursery pigs fed diets with or without mycotoxins, and with or without feed additive.

Item	Treatment ^1^					*p* Value		
	NC	PC	BG1	BG2	SEM	NC vs. PC	Linear (BG)	NC vs. (BG1 + BG2)
Jejunal mucosa, mg of protein								
TNF-α ^2^, pg	6.53	5.49	6.42	7.11	0.62	0.222	0.050	0.751
IL-6 ^3^, pg	14.38	17.97	15.34	20.14	3.27	0.391	0.597	0.372
MDA ^4^, nmol	0.31	0.25	0.28	0.34	0.07	0.354	0.159	0.992
PC ^5^, nmol	4.08	4.78	4.16	3.82	0.58	0.311	0.121	0.873
IgA ^6^, μg	6.89	6.29	6.60	3.23	1.35	0.709	0.060	0.170
IgG ^7^, μg	0.50	0.50	0.43	0.51	0.07	0.983	0.860	0.692

^1^ NC: control diet formulated with conventional corn; PC: inclusion of 8 mg/kg fumonisins and 0.19 mg/kg aflatoxins supplemented from mycotoxin-contaminated corn; BG1: PC diet with a feed additive (BG-Max, Church and Dwight Co., Inc., Princeton, NJ, USA) at 0.2%; BG2: PC diet with the feed additive at 0.4%; ^2^ TNF-α, tumor necrosis factor alpha; ^3^ IL-6, interleukin-6; ^4^ MDA, malondialdehyde; ^5^ PC, protein carbonyl; ^6^ IgA, immunoglobulin A; ^7^ IgG, immunoglobulin G.

**Table 5 toxins-15-00433-t005:** Intestinal morphology and crypt cell proliferation of nursery pigs fed diets with or without mycotoxins, and with or without feed additive.

Item	Treatment ^1^					*p* Value		
	NC	PC	BG1	BG2	SEM	NC vs. PC	Linear (BG)	NC vs. (BG1 + BG2)
Villus height, μm	397	430	440	451	21	0.263	0.492	0.066
Crypt depth, μm	270	250	259	248	11	0.189	0.891	0.218
VH:CD ^2^	1.51	1.59	1.71	1.85	0.13	0.621	0.199	0.098
Ki-67, % ^3^	38.7	38.6	39.4	39.2	1.3	0.980	0.705	0.706

^1^ NC: control diet formulated with conventional corn; PC: inclusion of 8 mg/kg fumonisins and 0.19 mg/kg aflatoxins supplemented from mycotoxin-contaminated corn; BG1: PC diet with a feed additive (BG-Max, Church and Dwight Co., Inc., Princeton, NJ, USA) at 0.2%; BG2: PC diet with the feed additive at 0.4%; ^2^ VH: CD, villus height-to-crypt depth ratio; ^3^ Ki-67, crypt cell proliferation rate.

**Table 6 toxins-15-00433-t006:** Composition of experimental diets.

Item	Phase 1		Phase 2		Phase 3	
	NC ^1^	PC ^1^	NC	PC	NC	PC
Ingredient, %						
Corn, yellow dent	32.04	19.54	46.30	33.80	68.45	55.95
Contaminated corn	0.00	12.50	0.00	12.50	0.00	12.50
Whey permeate	20.00	20.00	14.00	14.00	0.00	0.00
Soybean meal, dehulled	19.00	19.00	23.00	23.00	26.00	26.00
Cookie meal	10.00	10.00	5.00	5.00	0.00	0.00
Poultry meal	7.00	7.00	3.00	3.00	0.00	0.00
Fish meal	4.00	4.00	2.00	2.00	0.00	0.00
Blood plasma	4.00	4.00	2.00	2.00	0.00	0.00
Poultry fat	2.00	2.00	2.00	2.00	2.00	2.00
L-Lys HCl	0.50	0.50	0.50	0.50	0.54	0.54
L-Met	0.24	0.24	0.20	0.20	0.18	0.18
L-Thr	0.16	0.16	0.16	0.16	0.18	0.18
L-Trp	0.02	0.02	0.02	0.02	0.02	0.02
L-Val	0.00	0.00	0.04	0.04	0.08	0.08
Dicalcium phosphate	0.02	0.02	0.64	0.64	1.30	1.30
Limestone, ground	0.62	0.62	0.74	0.74	0.85	0.85
Salt	0.22	0.22	0.22	0.22	0.22	0.22
Vitamin premix ^2^	0.03	0.03	0.03	0.03	0.03	0.03
Mineral premix ^3^	0.15	0.15	0.15	0.15	0.15	0.15
Calculated composition, as-is						
Dry matter, %	91.0	91.0	90.3	90.3	89.4	89.4
ME, kcal/kg	3484	3484	3425	3425	3388	3388
Crude protein, %	24.2	24.2	22.2	22.2	18.9	18.9
SID ^4^ Lys, %	1.51	1.51	1.36	1.36	1.24	1.24
SID Met + Cys, %	0.84	0.84	0.75	0.75	0.69	0.69
SID Trp, %	0.26	0.26	0.24	0.24	0.21	0.21
SID Thr, %	0.88	0.88	0.80	0.80	0.74	0.74
Ca, %	0.86	0.86	0.80	0.80	0.72	0.72
STTD ^5^ P, %	0.46	0.46	0.41	0.41	0.34	0.34
Total P, %	0.68	0.68	0.64	0.64	0.60	0.60
Analyzed composition, as-is						
Dry matter, %	90.5	90.3	89.2	89.1	87.3	87.1
Crude protein, %	22.7	24.0	20.1	20.6	18.2	18.0
Crude ash, %	6.55	6.53	5.77	5.88	4.70	5.03
Ca, %	0.93	0.90	0.84	0.84	0.86	0.88
Total P, %	0.66	0.68	0.63	0.62	0.61	0.62
Aflatoxin, mg/kg	0.00	0.24	0.00	0.19	0.00	0.22
	(0.00) ^6^	(0.02)	(0.00)	(0.01)	(0.00)	(0.02)
Fumonisin, mg/kg	0.77	8.10	1.31	6.93	1.47	7.49
	(0.01)	(0.31)	(0.00)	(0.11)	(0.01)	(0.33)

^1^ NC: control diet formulated with conventional corn; PC: inclusion of 8 mg/kg fumonisins and 0.19 mg/kg aflatoxins supplemented from mycotoxin-contaminated corn; BG1: PC diet with a feed additive (BG-Max^TM^, Church and Dwight Co., Inc., Princeton, NJ, USA) at 0.2%; BG2: PC diet with the feed additive at 0.4%; ^2^ The vitamin premix provided per kilogram of complete diet: 6614 IU of vitamin A as vitamin A acetate, 992 IU of vitamin D3, 19.8 IU of vitamin E, 2.64 mg of vitamin K as menadione sodium bisulfate, 0.03 mg of vitamin B12, 4.63 mg of riboflavin, 18.52 mg of D-pantothenic acid as calcium pantothenate, 24.96 mg of niacin, and 0.07 mg of biotin; ^3^ The trace mineral premix provided per kilogram of complete diet: 33 mg of Mn as manganous oxide, 110 mg of Fe as ferrous sulfate, 110 mg of Zn as zinc sulfate, 16.5 mg of Cu as copper sulfate, 0.30 mg of I as ethylenediamine dihydroiodide, and 0.30 mg of Se as sodium selenite; ^4^ SID, standardized ileal digestibility; ^5^ STTD P, standardized total tract digestible phosphorus; ^6^ Numbers in parenthesis represent standard error.

**Table 7 toxins-15-00433-t007:** Sequence of primers for tight junction proteins in the jejunum of nursery pigs.

Gene ^1^	Primer Sequences (5′–3′) ^2^	Product Size, bp	A_T_ ^3^, °C	Reference
ZO-1	F: CAGAGACCAAGAGCCGTCC	105	60	Zhang et al. [70]
	R: TGCTTCAAGACATGGTTGGC			
OC	F: TCAGGTGCACCCTCCAGATT	118	60	Zhang et al. [70]
	R: AGGAGGTGGACTTTCAAGAGG			
CL-1	F: ATTTCAGGTCTGGCTATCTTAGTTGC	214	60	Zhang et al. [70]
	R: AGGGCCTTGGTGTTGGGTAA			
GAPDH	F: TCGGAGTGAACGGATTTGGC	147	60	Chen et al. [71]
	R: TGCCGTGGGTGGAATCATAC			

^1^ ZO-1, zonula occludens; OC, occludin; CL-1, claudin-1; GAPDH, glyceraldehyde-3-phosphate dehydrogenase; ^2^ F, forward; R, reverse; ^3^ A_T_, annealing temperature.

## Data Availability

The data presented in this study are available in this article.

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
