# Peer review of "Efficacy of Feed Additive Containing Bentonite and Enzymatically Hydrolyzed Yeast on Intestinal Health and Growth of Newly Weaned Pigs under Chronic Dietary Challenges of Fumonisin and Aflatoxin"

_toxins, 2023, doi:10.3390/toxins15070433_

Round 1
Reviewer 1 Report
The authors' research in this article shows that the addition of bentonite and enzymatic yeast (EHY) to pig feed can reduce the risk of mycotoxins and improve pig growth performance. The results show that the additive positively modulates the immune response and enhances intestinal integrity and morphology, leading to better growth performance in newly weaned pigs. Further research should be required in the article on how feed additives containing bentonite and EHY affect the gut microbiota under the challenge of FUM and AF to fully explain the immune response. Articles should pay attention to grammar.
3 Discussion When feeding multiple mycotoxins, the mycotoxins may have synergistic toxic effects and may cause additional damage to growth performance, and the consequences of harm caused by synergistic toxicity can be briefly enumerated.
3 Discussion The expression of dense protein-1 (CL-1) in this study increased linearly with increasing feed additive supplementation, and charts could be added for expression.
3 Discussion 275. The writer tries not to use indeterminate inflections such as speculation and possible, and can give examples for verification.
5.2 Is the freezing on during centrifugation and are there any temperature requirements.
5.2 Steps and considerations when collecting blood samples.
5.4 Configuration of phosphate buffered saline (PBS), what is the pH value.
The authors should pay attention to the number of the title.
Author Response
The authors' research in this article shows that the addition of bentonite and enzymatic yeast (EHY) to pig feed can reduce the risk of mycotoxins and improve pig growth performance. The results show that the additive positively modulates the immune response and enhances intestinal integrity and morphology, leading to better growth performance in newly weaned pigs. Further research should be required in the article on how feed additives containing bentonite and EHY affect the gut microbiota under the challenge of FUM and AF to fully explain the immune response. Articles should pay attention to grammar.
= Thank you for providing thorough and valuable comments for our manuscript. We agree with that microbiota result could be strong evidence to explain the immune response and it would be our further step to investigate the potential mechanism that mycotoxin affect immune response. We considered all comments from the Reviewer #1 and made the needed revision in the manuscript.
3 Discussion When feeding multiple mycotoxins, the mycotoxins may have synergistic toxic effects and may cause additional damage to growth performance, and the consequences of harm caused by synergistic toxicity can be briefly enumerated.
= L166-169: Thank you for your comment. We newly added some examples about synergistic toxicity. Now the text reads as: ' Furthermore, previous studies have also indicated that mycotoxins may have synergistic toxic effects, such as impaired immune health, systemic inflammation, and hepatic histopathology changes, that can cause more damages on the growth performance when multiple mycotoxins are fed [12,29,30].'
3 Discussion The expression of dense protein-1 (CL-1) in this study increased linearly with increasing feed additive supplementation, and charts could be added for expression.
= Figure 1: Thank you for your comment. We agreed to that charts could be added for better expression. We changed table 5 to figure 1 to present our gene expression result.
Figure 1. Tight junction proteins of jejunum of nursery pigs fed diets with or without mycotoxins, and with or without feed additive. (A) Relative expression of zonula occludens in jejunum of pigs. (B) Relative expression of occludin in jejunum of pigs. (C) Relative expression of claudin-1 in jejunum of pigs.
3 Discussion 275. The writer tries not to use indeterminate inflections such as speculation and possible, and can give examples for verification.
= L 261-266: Thank you for your comment. For the improved tight junction protein expression, we tried to explain this result in two reasons. First, the adsorption ability from feed additive. Second, the functional compounds from yeast that can stimulate tight junction protein expression. In addition, we agree that indeterminate inflections should be avoided. We changed the word 'might' to 'could' to avoid ambiguity.
5.2 Is the freezing on during centrifugation and are there any temperature requirements.
= L342: Thank you for your question. Yes, we kept the centrifuge at 4 °C when we processed our samples. We newly added the statement to describe this. Now the text reads as: '... the serum samples were centrifuge at 3,000 × g on 4 °C for 15 min...'
5.2 Steps and considerations when collecting blood samples.
= L338-347: Thank you for your comment. We newly added more details about blood collection. Now the text reads as: ' On d 28, pigs were placed in a recumbent position on a V shaped table to restrict their movement and blood samples of each pig were collected (10 mL) using vacutainer tubes (Becton Dickinson Vacutainer Systems, Franklin Lakes, NJ, USA) and needles (0.8mm×32mm needles, Eclipse, Becton Dickinson Vacutainer Systems) by puncturing the jugular vein……. In order to prevent prolonged stress from blood sampling, the blood samples were collected 4 days before to the end of the feeding period.'
5.4 Configuration of phosphate buffered saline (PBS), what is the pH value.
L392-383: Thank you for your comment. We newly added more details about PBS. Now the text reads as: '...1 mL of phosphate-buffered saline (PBS, 0.01M phosphate, 0.0027M KCl, 0.137M NaCl, pH 7.4 at 25°C).'
The authors should pay attention to the number of the title.
= Thank you for your comment. We have rechecked the entire manuscript for the number of the titles.

Reviewer 2 Report
Dear author(s),
there are some inspiring insights thorough the manuscript and I tend to agree on its publication. However, there are few points that needs to be quickly addressed to improve its overall communication:
Title:
1/ significant shortening advisable, condensate the main doscovery into a short and groundbreaking claim
Abstract:
2/ strictly follow the established schema of writing academic Abstract: A/ introduction (urgency and significance of the research hypothesis); B/ principles of the methods used + key results; C/ conclusions (commercial and environmental impacts)
3/ reduce the use of abbreviations, jargon and technical terms, please understand that the purpose of the Abstract is to explain to all readers (including those from other disciplines) what the paper is about
4/ there is no reason to go into detail and present the results obtained under specific reaction conditions, rather provide a synthesis of the results obtained
5/ quantify the industrial importance of your work in financial terms
Introduction:
6/ remove all clusters of references to avoid reference overkill (prefer only 1 reference to support 1 claim)
7/ complexity of phosphorus availability to organisms should be better explained, refer to Fig. 1 in paper "Novel sorbent shows promising financial results on P recovery from sludge water"
8/ make sure that this chapter fully introduces any reader into to the topic, explain all the terms, units, abbreviations, Latin and Greek letters, and the whole context that is necessary for anyone (including experts from other disciplines) to understand the following chapters
9/ better explain the economic complexity (refer to papers "The Dynamic Effect of Micro-Structural Shocks on Private Investment Behavior" and "The analysis of investment into industries based on portfolio managers")
10/ the research hypothesis could be stated more clearly, condensate the research hypothesis into 1 short statement (or question) that will be subsequently confirmed or refuted, make sure the urgency and significance of the research hypothesis was justified in its environmental - economic nexus
Results:
11/ each Tab. and Fig. should be provided with caption that describes A/ what can be seen and B/ how is this relevant to the research hypothesis
12/ avoid data overkill, present only the most most industrially important results with a preference for those that are easier to interpret economically
13/ provide cost breakdown or at least some simplified financial analysis
Discussion:
14/ show more self-criticism to your work (is this a representative sample? can all the methods and results be fully trusted? what are the weaknesses of the methods used? where do the main measurement inaccuracies arise? what are the limitations from a commercial point of view? are the lessons learned transferable to other fields?)
15/ propose some improvements (such as use of biochar or BSF - refer to papers "Environmental and economic advantages of production and application of digestate biochar" and "Insect rearing on biowaste represents a competitive advantage for fish farming") and directions for future research
Materials and Methods:
16/ the method must be presented in such a way that it can be reproduced anytime, by anyone, anywhere, please understand that the methodology must be described in a completely unambiguous way that does not allow for multiple interpretations (everyone who reads this chapter should get very precise instructions on how to repeat your procedure to achieve exactly the same results)
17/ each material/reactant and apparatus used needs to be presented in detail (serial number, setup, process parameters, manufacturer, country of origin, purity etc.)
18/ provide cost breakdown or at least some simplified financial analysis if you are about to argue that this concept is realistic
Author Response
Dear Reviewer, We greatly appreicate your review of the manuscript. Could you submit your review comment again as I may think that the comment was for a different manuscript (and this happens as we review so many papers!).
Sincerely, Sung Woo Kim (Corresponding Author)
Reviewer 3 Report
The submitted article describes the results of an experiment which compared the actions of two concentrations of a commercial feed additive to a positive and negative control. There is an extensive literature on bentonite in swine diets so was surprised by the lack of effect in this study. In particular, the SA:SO ratios were not different from the positive control. In general the lack of statistical significance (P < 0.05) was disappointing. Things are either significant or not. The article was well written and I did enjoy reading it.
Author Response
The submitted article describes the results of an experiment which compared the actions of two concentrations of a commercial feed additive to a positive and negative control. There is an extensive literature on bentonite in swine diets so was surprised by the lack of effect in this study. In particular, the SA:SO ratios were not different from the positive control. In general the lack of statistical significance (P < 0.05) was disappointing. Things are either significant or not. The article was well written and I did enjoy reading it.
= Thank you for providing thorough and valuable comments for our manuscript. Part of results in current manuscript are interesting because some of them are not consistent with previous studies. But there were meaningful findings in current manuscript that supported our hypothesis, which brought the highlights of this paper.

Round 2
Reviewer 1 Report
Pay attention to format and language problems, try to refer to the literature in the past two years.
It doesn't have to be complicated. Simple sentences are recommended.
Author Response
Dear Reviewer, Thank you again for further comments on the manuscript. The authors went through another round of revision and made minor editorial changes in the text.